# REAL-TIME UNCERTAINTY DECOMPOSITION FOR ONLINE LEARNING CONTROL

## ABSTRACT

Safety-critical decisions based on machine learning models require a clear understanding of the involved uncertainties to avoid hazardous or risky situations. While aleatoric uncertainty can be explicitly modeled given a parametric description, epistemic uncertainty rather describes the presence or absence of training data. This paper proposes a novel generic method for modeling epistemic uncertainty and shows its advantages over existing approaches for neural networks on various data sets. It can be directly combined with aleatoric uncertainty estimates and allows for prediction in real-time as the inference is sample-free. We exploit this property in a model-based quadcopter control setting and demonstrate how the controller benefits from a differentiation between aleatoric and epistemic uncertainty in online learning of thermal disturbances.

## 1 INTRODUCTION

With improved sensor quality and more powerful computational resources, data-driven models are increasingly applied in safety-critical domains such as autonomous driving or human-robot interaction (Grigorescu et al., 2020). However, measurements usually suffer from noise and the available data is often scarce compared to all possible states of a complex environment. This requires controllers, which rely on supervised learning techniques, to properly react to ignorance and imprecision in the model to avoid dangerous situations. In order to allow an implementation of risk-averse (for exploitation and safety improvements) or risk-seeking (for exploration) behavior, the model must clearly disaggregate the information in the data into more than just the "best estimate" and differentiate between different sources of uncertainty. Besides the point estimate of a model, one can distinguish aleatoric (uncertainty in the data) and epistemic (uncertainty in the model) uncertainty. The former is irreducible as it is inherent to the stochastic process the data is recorded from, while the latter origins from a limited expressive power of the model or scarce training samples (Der Kiureghian & Ditlevsen, 2009).

Gaussian processes (GPs) inherently provide a measure for its fidelity with the posterior standard deviation prediction (Rasmussen & Williams, 2006). It also allows to differentiate aleatoric uncertainty (typically considered as observation noise) and epistemic uncertainty (modeled by the kernel). However, the former allows only homoscedastic (constant) estimates, while real-world applications typically require heteroscedastic uncertainty models. An extension to heteroscedastic GP regression is presented in (Lazaro-Gredilla & Titsias, 2011), however, it is a variational approximation and further increases the computational complexity and GPs generally suffer from poor scaling to large datasets (Quinonero-Candela & Rasmussen, 2005).

In deep learning, the modeling of uncertainties also gained increasing interest over the past years (Kendall & Gal, 2017). Heteroscedastic aleatoric uncertainty can be captured well, if the output of the stochastic process can directly be observed and its parametric distribution is known. However, for more general cases, approximation techniques such as variational inference or sampling is required (Bishop, 2006). For epistemic uncertainty estimation with neural networks (NN), the key idea for most approaches can be summarized as follows. Randomness is introduced to the neural network through sampling during training and inference. While the training robustifies the network against the injected noise at the training locations, it allows the noise to pass to the output at input locations where no training data is available. For inference, multiple predictions of the network are sampled for the same inputs, allowing to compute a statistical measure for the uncertainty

at the output (Depeweg et al., 2018; Depeweg, 2019). However, sampling the network during inference is a high computational burden, and is therefore not suitable in real-time critical control tasks. An ensemble based approach by (Lakshminarayanan et al., 2017) works with far less instances of a network, but does not differentiate between aleatoric and epistemic uncertainty explicitly.

Despite those drawbacks in the uncertainty representation of data-driven models, the control community started to incorporate them increasingly in the decision making for various applications. For example Fanger et al. (2016) uses an epistemic uncertainty measure to dynamically assign leader order follower roles for cooperative robotic manipulation. The work by Berkenkamp et al. (2016) ensures a safe exploration of an unknown task space based on GP error bounds and a gain scheduling approach for computed torque control is presented in Beckers et al. (2019). The work by Liu et al. (2020) considers the epistemic uncertainty as an estimate of the distance between source and target domains (known as domain shift) to design a robust controller. In Umlauft & Hirche (2020) and Chowdhary et al. (2015) an online learning control approach for GPs models is considered, which approach the dual control problem (Wittenmark, 1995) as a model-based adaptive control problem. The work by Yesildirak & Lewis (1995) uses neural network for adaptive control in a continuous time fashion, which relies on a time-triggered (periodic) update of the model rather than a event-based adaptation as we propose in this work. More general, risk averse control strategies have been presented by Umlauft et al. (2018); Medina et al. (2013); Todorov & Li (2005). However, all of these approaches only consider the model fidelity in general and do not differentiate between aleatoric and epistemic uncertainty.

Therefore, the main contributions of this paper are the following. We propose a deep learning framework with a real-time capable epistemic uncertainty prediction. The resulting online learning model is employed by a controller, which shows a distinct reaction to epistemic and aleatoric uncertainty. We evaluate the proposed methods on synthetic and real-world benchmark data sets, and simulate a quadcopter controller, which learns online the disturbances injected by thermals.

## 2 PROBLEM FORMULATION

Consider the discrete-time dynamical system[1] with control $\boldsymbol{u} \in \mathbb{U} \subseteq \mathbb{R}^{d_u}$ and state $\boldsymbol{x} \in \mathbb{X} \subseteq \mathbb{R}^{d_x}$

$$\boldsymbol{x}_{k+1} = \boldsymbol{g}(\boldsymbol{x}_k, \boldsymbol{u}_k) + \boldsymbol{y}_k, \tag{1}$$

where $\boldsymbol{g} \colon \mathbb{X} \times \mathbb{U} \to \mathbb{X}$ is known, while $\boldsymbol{y}$ is a i.i.d. random vector sampled in every time step from

$$\boldsymbol{y}_k \sim \mathcal{D}(\boldsymbol{f}(\boldsymbol{x}_k)), \tag{2}$$

where $\mathcal{D}(\cdot)$ denotes a known distribution over real vectors $\boldsymbol{y} \in \mathbb{Y} \subseteq \mathbb{R}^{d_x}$ and depends on the parameters $\boldsymbol{p} \in \mathbb{P} \subseteq \mathbb{R}^{d_p}$. These state-dependent parameters arise from an unknown mapping $\boldsymbol{f} \colon \mathbb{X} \to \mathbb{P}$. We denote the unknown component $\boldsymbol{y}_k$ of the dynamical system generally as disturbance but it could also be the unmodeled part of the dynamics, such as friction or serve as black-box model for the dynamics if no analytic description is available ($\boldsymbol{g}(\cdot, \cdot) = 0$). We assume measurements can be taken to obtain the data set $\mathbb{D}_{\text{tr}} = \{(\boldsymbol{x}_i, \boldsymbol{y}_i)\}_{i=1}^{N_{\text{tr}}}$ with inputs $\mathbb{X}_{\text{tr}} = \{\boldsymbol{x}_i\}_{i=1}^{N_{\text{tr}}}$ and outputs $\mathbb{Y}_{\text{tr}} = \{\boldsymbol{y}_i\}_{i=1}^{N_{\text{tr}}}$, such that a model $\hat{\boldsymbol{f}}(\cdot)$ of $\boldsymbol{f}(\cdot)$ can be learned. $N_{\text{tr}} \in \mathbb{N}$ denotes the current number of training data points and is initially zero, i.e., the training set is empty.

The task is to choose a control input $\boldsymbol{u}_k$, such that the system (1) follows a given reference $\boldsymbol{x}^{\text{des}}$. Furthermore, the controller can take new measurements of $\boldsymbol{y}$ to improve its model over time. We consider each measurement of $\boldsymbol{y}$ to be costly and therefore new training points should only be collected when necessary. Applications, where data collection is costly can be found in distributed systems, where multiple sensors share the same scarce communication channel, or in autonomous systems with limited data storage capacity.

The need for high data efficiency requires models, which judge upon their own fidelity in real-time to identify valuable measurements. As existing approaches for modeling epistemic uncertainty in deep learning suffer from a high computational complexity we first focus on developing a novel method for epistemic uncertainty predictions before proposing an online learning control strategy which makes use of a neural network model decomposing its uncertainties.

---

[1]Bold/capital symbols generally denote vectors/matrices, $\mathcal{D}(\cdot)/\mathcal{U}(\cdot)/\mathcal{N}(\cdot)/\mathcal{B}(\cdot)$ a general parametric/the uniform/Gaussian/Bernoulli distribution, respectively.

## 3 EPISTEMIC UNCERTAINTY ESTIMATION

### 3.1 RELATED WORK

Learning an epistemic uncertainty estimator is not straight forward as it measures the absence of training data. Most prominently Gaussian processes with stationary kernels offer such a measure implicitly with their posterior variance prediction. However, GPs are known to scale poorly for large data sets: While regression and uncertainty predictions can be performed with $\mathcal{O}(N_{\text{tr}})$ and $\mathcal{O}(N_{\text{tr}}^2)$, respectively, considering a new data point takes $\mathcal{O}(N_{\text{tr}}^3)$ computations (also without hyperparameter optimization, $\mathcal{O}(N_{\text{tr}}^2)$ for rank-1 update methods). While various methods have been proposed to make GP computationally more efficient, including sparse GPs (Quinonero-Candela & Rasmussen, 2005), distributed GPs (Deisenroth & Ng, 2015) and local GPs (Nguyen-Tuong et al., 2009a;b), these approximations typically focus only on the precision of the point estimate and distort the uncertainty prediction. For estimating the "distance" to the training set, kernel density estimation (KDE) can also be used (Rosenblatt, 1956), however, the non-parametric nature implies that the inference time grows linearly with the number of considered data points, which we aim to avoid.

More recently, several different approaches for epistemic uncertainty estimates using deep learning frameworks have been proposed. Popular approaches rely on Bayesian approximations(Depeweg et al., 2016) or permanent dropouts (not only during training to avoid overfitting) (Gal, 2016; Gal & Ghahramani, 2016). Furthermore, latent inputs can also be used to achieve a decomposition into aleatoric and epistemic uncertainty as presented in (Depeweg et al., 2017). However, in particular for Bayesian NNs, these approaches become computationally challenging. Firstly, they have a larger number of parameters to tune than their deterministic counterparts and rely on variational inference methods (Kwon et al., 2020). Secondly, the prediction requires to sample the entire network before the statistics of the output can be computed. For the application in real-time critical control problems (e.g., robotics with a sampling rate of $1\,\text{kHz}$), these computational burdens prohibit an employment of these techniques. A sampling-free estimation methods is proposed by Postels et al. (2019), but suffers from a quadratic space complexity in the number of weights in the network and relies on first-order Taylor approximations in the propagation of the uncertainties, which might become inaccurate depending on the non-linearity of the activation functions

### 3.2 *EpiOut* - EXPLICITLY LEARNING EPISTEMIC UNCERTAINTY

In order to allow the estimation of epistemic uncertainty in real-time, we introduce the idea of explicitly modeling it with a separate output of a neural network, calling it *EpiOut*. Since the epistemic uncertainty expresses the absence of data, the original data set $\mathbb{D}_{\text{tr}}$ does not contain data for training *EpiOut*. Therefore, we generate an epistemic uncertainty data set, with inputs $\mathbb{X}_{\text{epi}} = \{\tilde{\boldsymbol{x}}_j\}_{j=1}^{N_{\text{epi}}}$ and outputs $\mathbb{Y}_{\text{epi}} = \{\tilde{y}_j\}_{j=1}^{N_{\text{epi}}}$ concatenated in $\mathbb{D}_{\text{epi}} = \{(\tilde{\boldsymbol{x}}_j, \tilde{y}_j)\}_{j=1}^{N_{\text{epi}}}, N_{\text{epi}} \in \mathbb{N}$.

Different variations for sampling the set $\mathbb{X}_{\text{epi}}$ can be chosen depending on the desired scalability properties. A naive approach is to sample the entire input space uniformly, which suffers from the curse of dimensionality. Alternatively, we propose to sample around existing training points from a normal distribution

$$\mathbb{X}_{\text{epi}} = \bigcup_{i=1}^{N_{\text{tr}}} \{\tilde{\boldsymbol{x}}_j \sim \mathcal{N}(\boldsymbol{x}_i, \Gamma), j = 1, \dots, N_{\text{epi}}/N_{\text{tr}}\}, \tag{3}$$

where we implicitly assume that $N_{\text{epi}}$ is chosen such that $\delta = N_{\text{epi}}/N_{\text{tr}}$ is a positive integer. Supposing that a standardization of the input space to unity is performed based on $\mathbb{X}_{\text{tr}}$, $\Gamma = \boldsymbol{I}$ can be chosen if no further knowledge on $\boldsymbol{f}(\cdot)$ is available. Otherwise, scaling $\Gamma$ can be interpreted similarly to the lengthscale of a GP as a measure for how far away from a training point the prediction is reliable: Larger $\Gamma$ will lead to further spread of $\mathbb{X}_{\text{epi}}$ and therefore low epistemic uncertainty in the neighborhood of the training data, which would be meaningful if the true function is known to have a low Lipschitz constant, and vice versa.

We propose to set $\delta$ to a multiple of $2d_x + 1$ which corresponds to the intuition to pad each training point in both directions of each dimension with a *epi point* $\tilde{\boldsymbol{x}}$. The reasoning behind the additional $+1$ point will become clear in the following. To define the set $\mathbb{Y}_{\text{epi}}$, we first compute the minimal

distance (according to any distance metric $d\colon \mathbb{X} \times \mathbb{X} \to \mathbb{R}_{0,+}$) to the training data for each epi point

$$d_j = \min_{\boldsymbol{x} \in \mathbb{X}_{\mathrm{tr}}} d(\tilde{\boldsymbol{x}}_j, \boldsymbol{x}), \quad j = 1, \dots, N_{\mathrm{epi}}, \tag{4}$$

keeping in mind that the closest training data point is not necessarily the one used to generate the sample. Let $d_{N_{\mathrm{tr}}}$ be the $N_{\mathrm{tr}}$-th smallest element of all $d_j$, we generate $\mathbb{Y}_{\mathrm{epi}}$ and update $\mathbb{X}_{\mathrm{epi}}$ as follows

$$\tilde{y}_j = \begin{cases} 1, & \text{if } d_j > d_{N_{\mathrm{tr}}} \\ 0, & \tilde{\boldsymbol{x}}_j \leftarrow \arg\min_{\boldsymbol{x} \in \mathbb{X}_{\mathrm{tr}}} d(\tilde{\boldsymbol{x}}_j, \boldsymbol{x}) \quad \text{if } d_j \leq d_{N_{\mathrm{tr}}} \end{cases}. \tag{5}$$

Thus, the $N_{\mathrm{tr}}$ points in $\mathbb{X}_{\mathrm{epi}}$ with the least distance to a training point are replaced by the corresponding point in $\mathbb{X}_{\mathrm{tr}}$. Now the choice of $2d_x + 1$ epi points becomes clear as one of them will be turned into $\tilde{y} = 0$ corresponding to "low epistemic uncertainty", while $2d_x$ points further away from the training point $\tilde{y} = 0$ indicate the opposite.

Given the data set $\mathbb{D}_{\mathrm{epi}}$, the neural network is now equipped with one additional output, i.e., the parameter layer is $d_p + 1$ dimensional with output $[\hat{\boldsymbol{f}}(\cdot)\ \eta(\cdot)]^T$. The new output $\eta(\cdot)$ is terminated with a neuron using a sigmoidal activation function, such that $\eta\colon \mathbb{X} \to [0, 1]$. This is beneficial because it allows immediately to judge, whether the predicted uncertainty is *high* ($\approx 1$) or *low* ($\approx 0$) without any reference evaluation (see comparison to alternanative methods below).

Independently of the loss function for the original network, the augmented output, also considered as *epistemic output* is trained using a binary cross-entropy loss, which is the natural choice for binary outputs. It quantifies the uncertainty in the prediction of the other outputs based on the distance to the training data measured by $d(\cdot, \cdot)$. For the sake of focus, we will be using the Euclidean distance, however the method can be easily extended to other metrics and we leave it to future work to investigate alternatives.

### 3.3 Computational complexity

The analysis of the computational complexity shows that (3) is a $\mathcal{O}(N_{\mathrm{epi}}) \mathrel{\hat{=}} \mathcal{O}(N_{\mathrm{tr}} d_x)$ operation, whereas (4) is for a trivial implementation a $\mathcal{O}(N_{\mathrm{tr}} N_{\mathrm{epi}}) \mathrel{\hat{=}} \mathcal{O}(d_x N_{\mathrm{tr}}^2)$ operation. However, an implementation based on kd-tree (Cormen et al., 2009) allows an execution in $\mathcal{O}(N_{\mathrm{epi}} \log(N_{\mathrm{epi}})) \mathrel{\hat{=}} \mathcal{O}(d_x N_{\mathrm{tr}} \log(N_{\mathrm{tr}} d_x))$ time. Finding the $N_{\mathrm{tr}}$ smallest distances from all $N_{\mathrm{epi}}$ points in (5) can obtained in $\mathcal{O}(N_{\mathrm{tr}} + (N_{\mathrm{epi}} - N_{\mathrm{tr}}) \log(N_{\mathrm{tr}})) \mathrel{\hat{=}} \mathcal{O}(N_{\mathrm{tr}} + N_{\mathrm{tr}}(d_x - 1) \log(Ntr))$ time. The training of neural network with a fixed number of weights requires $\mathcal{O}(N_{\mathrm{epi}}) \mathrel{\hat{=}} \mathcal{O}(N_{\mathrm{tr}} d_x)$. Hence, the overall complexity results in $\mathcal{O}(d_x N_{\mathrm{tr}} \log(d_x N_{\mathrm{tr}}))$, and it is straight forward to derive an overall space complexity of $\mathcal{O}(N_{\mathrm{epi}} d_x) \mathrel{\hat{=}} \mathcal{O}(N_{\mathrm{tr}} d_x^2)$ for storing the set $\mathbb{X}_{\mathrm{epi}}$. The following should be considered when comparing to classical deep learning frameworks which generally can be trained in linear time.

- When used on streaming data (as for online learning control), the set $\mathbb{D}_{\mathrm{epi}}$ can be constructed iteratively, reducing the complexity to $\mathcal{O}(\log(N_{\mathrm{tr}}))$
- The most time critical computation (4) can efficiently be parallelized on a GPU.
- The method is designed for applications where measuring data is considered costly and therefore sparse data can be expected.

### 3.4 Evaluation

For evaluation we compare the following models. The implementation is available in the supplementary material.

- vanilla *GP model* with a squared exponential automatic relevance determination kernel based on the GPy implementation[2]
- *BNN*: Bayesian Neural Network with 2 fully connected hidden layers each with 50 hidden units and normal distributions over their weights based on this implementation.[3]

---

[2] https://sheffieldml.github.io/GPy/
[3] https://matthewmcateer.me/blog/a-quick-intro-to-bayesian-neural-networks/

- *Dropout*: A neural network with 2 fully connected permanent layers each with 50 hidden units with dropout probability $\rho = 0.05$. [4].
- *EpiOut*: The proposed model with 2 fully connected layers (50 neurons) and $\Gamma = \boldsymbol{I}$, $\delta = 2$.

For the evaluation we utilize a weighted mean square error measures defined as follows

$$\rho = \frac{\sum_{i=1}^{N_{\text{te}}} (y_i - \hat{f}(\boldsymbol{x}_i))^2 (1 - \eta(\boldsymbol{x}_i))}{\sum_{i=1}^{N_{\text{te}}} (1 - \eta(\boldsymbol{x}_i))}, \tag{6}$$

i.e., if the model decides that it is uncertain about the prediction at a test point, the squared error for this prediction is discounted (weighted less). However, the model can only achieve a low $\rho$ if it is also certain at some test points, because the denominator, goes to zero for many uncertain predictions. In consequence, $\rho$ is only defined if $\eta(\cdot) < 1$ holds for at least one test point. Furthermore, the total discount, defined as $\sum_{i=1}^{N} \eta(\boldsymbol{x}_i)$ can additionally be utilized for a plausibility check of the epistemic uncertainty predictions since it should generally be larger on the test than on the training data set.

The measure in (6) relies on epistemic uncertainty prediction in the interval $[0, 1]$. This is only ensured for the proposed *EpiOut* approach and therefore the uncertainty measures, more specifically the predicted standard deviation, provided by the GP, *Dropout* and *BNN* are scaled to the unit interval based on the evaluation on all test and training points.

The following data sets are utilized for evaluation.

- *1D Center*: The nominal function is $f(x) = \sin(x\pi)$, with training points $\mathbb{X}_{\text{tr}} = \{x_i \sim \mathcal{U}(-1, 1)\}_{i=1}^{100}$ and $N_{\text{te}} = 961$ test points are placed on a grid $[-4, 4]$.
- *1D Split*: Same as *1D Center*, but $\mathbb{X}_{\text{tr}} = \{x_i \sim \mathcal{U}(-2, -1)\}_{i=1}^{100} \cup \{x_i \sim \mathcal{U}(1, 2)\}_{i=101}^{200}$.
- *2D Gaussian*: The nominal function ($d_x = 2$, $d_p = 1$) is $f(\boldsymbol{x}) = \frac{\sin(5x_1)}{5x_1} + x_2^2$ with training points $\mathbb{X}_{\text{tr}} = \left\{\boldsymbol{x}_i \sim \mathcal{N}\left(\begin{bmatrix} -1 \\ 0 \end{bmatrix}, \begin{bmatrix} 0.02 & 0 \\ 0 & 0.1 \end{bmatrix}\right)\right\}_{i=1}^{500} \cup \left\{\boldsymbol{x}_i \sim \mathcal{N}\left(\begin{bmatrix} 1 \\ 0 \end{bmatrix}, \begin{bmatrix} 0.02 & 0 \\ 0 & 0.1 \end{bmatrix}\right)\right\}_{i=501}^{1000}$ and $N_{\text{te}} = 961$ test points are uniformly placed on a grid $[-2, 2]^2$.
- *2D Square*: Same as *2D Gaussian*, but with with $N_{\text{tr}} = 80$ training points placed uniformly along the boundary of the square $[-1, 1]^2$.
- *PMSM temperature* is a 2Hz recording ($d_x = 8$, $d_y = 1$) of the temperature from a permanent magnet synchronous motor.[5]. To allow a comparison with the GP within reasonable computational limits, $N_{\text{tr}} = 5000$ and $N_{\text{te}} = 1000$ points were randomly extracted from a total of $\approx 10^6$ samples.
- *Sarcos* is a data set for learning the inverse dynamics of a seven degrees-of-freedom SARCOS anthropomorphic robot arm $d_x = 21$, $d_p = 1$.[6]. $N_{\text{tr}} = 10000$ and $N_{\text{te}} = 2000$ points were randomly extracted from a total of $\approx 5 \times 10^4$ samples.

### 3.5 RESULTS & DISCUSSION

The numerical results are presented in Table 1 and an illustration for the data set *1D Split* for all models is shown in Fig. 1. Besides showing empirically an advantage over existing approaches we want to point out the following benefits.

- The *EpiOut* model predicts the uncertainty measure in a sample free manner. This is crucial in data-efficient online learning scenarios, where the epistemic uncertainty is used to evaluate the usefulness of an incoming data point to decide upon its rejection. Hence, it is called more frequently than the online training function and must be computationally efficient. The prediction time for *EpiOut* is typically an order of magnitude faster than *Dropout* and *BNN*.

---

[4]https://github.com/yaringal/DropoutUncertaintyDemos
[5]https://www.kaggle.com/wkirgsn/electric-motor-temperature
[6]http://www.gaussianprocess.org/gpml/data/

Table 1: Weighted mean squared error $\rho$ as defined in (6) for the considered models on different data sets. The GP model is grayed out since it does not scale towards larger data sets. To prevent the scaling of the epistemic uncertainty to the unit interval from deteriorating the performance of *GP model*, *BNN* and *Dropout* we minimize $\rho$ for the evaluation over a factor $\alpha \in [0, 1]$, $\tilde{\eta} = \alpha\eta$ to

|         | 1D Center | 1D Split | 2D Square | 2D Gaussian | PMSM temperature | Sarcos  |
|---------|-----------|----------|-----------|-------------|------------------|---------|
| GPmodel | 0.1660    | 0.1207   | 0.0655    | 0.0342      | 0.0007           | 4.8150  |
| BNN     | 1.4177    | 0.5889   | 1.1953    | 0.8830      | 0.0101           | 21.9015 |
| Dropout | 0.7904    | 0.6181   | 1.0681    | 0.4468      | 0.1490           | 21.3842 |
| EpiOut  | **0.0480**| **0.1125**| **0.1893**| **0.0056** | **0.0024**       | **14.2537** |

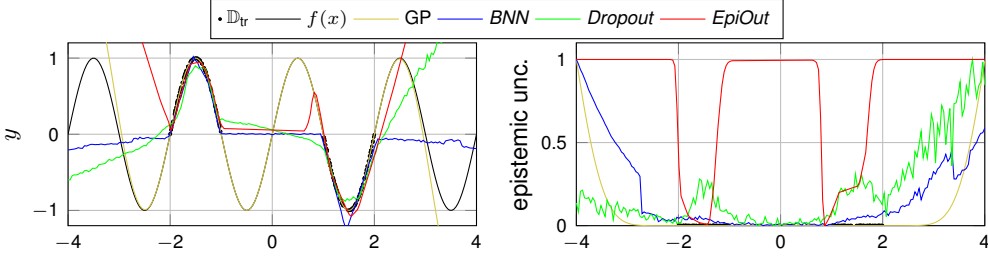

Figure 1: The point estimate (mean prediction) for the different models along with the training data *1D split* and the true underlying function $f(x) = \sin(\pi x)$ are shown on the left. The right plot shows the epistemic uncertainty estimate, where *BNN* and *Dropout* clearly miss to predict a higher uncertainty between the data clusters.

- A single evaluation of $\eta(\cdot)$ is sufficient for a conclusion whether the uncertainty is high or low, since it is bounded to the interval $[0, 1]$, whereas alternative approaches based *BNN* and *Dropout* provide a return value $[0, \infty]$, which can be difficult to interpret without a maximum value as reference.

For more extensive results and exact computation times, we refer to the supplementary material.

## 4 ONLINE LEARNING CONTROL USING UNCERTAINTY DECOMPOSITION

### 4.1 EVALUATION IN A QUADCOPTER CONTROL TASK

As an application of our proposed approach, we consider the task to control a quadcopter which explores a given terrain with unknown thermals[7]. We assume that the quadcopter dynamics i.e. $g(\cdot, \cdot)$ in 1 is known (compare (Shi et al., 2019)). The thermals act on the quadcopter as a disturbance on the acceleration in $z^q$-direction[8] We model the disturbance as normal distribution $\mathcal{N}(\mu(x), \sigma^2(x))$, leading to $p = [\mu, \sigma]^T$ and the NN models the state dependency of these parameters $\hat{f} : \mathbb{X} \to \mathbb{P}$. Initially, there is no training data available and the desired trajectory $x^{\text{des}}$ are three rounds at constant height $z^q = 0$ on a square in the $x^q$-$y^q$-plane with edge length $0.1$ followed by three rounds with edge length $0.05$.

The online learning control approach for achieving this task with a high tracking precision and data-efficiency is presented in the following.

### 4.2 LEARNING CONTROL DESIGN

The *EpiOut* approach is sufficiently fast to be implemented in a mobile real-time control loop and therefore serves as trigger when new data points must be measured. By sampling from a Bernoulli

---

[7]The data of the thermals is taken from publicly available paragliding data `https://thermal.kk7.ch`.
[8]Superscripts $q$ distinguish the quadcopter position $x^q, y^q, z^q$ from the general inputs $x$ and outputs $y$.

distribution, whose parameter corresponds to $\eta(\cdot)$, new measurements $(\boldsymbol{x}_i, \boldsymbol{y}_i)$ are added to the training according to

$$\mathbb{D}_{\mathrm{tr}} \leftarrow \mathbb{D}_{\mathrm{tr}} \cup \begin{cases} (\boldsymbol{x}_i, \boldsymbol{y}_i) & \text{if i} = 1 \\ \emptyset & \text{if i} = 0 \end{cases} \qquad \text{where i} \sim \mathcal{B}(\alpha), \quad \alpha = \eta(\boldsymbol{x}_i). \tag{7}$$

This ensures a high accuracy of the disturbance model $\hat{\boldsymbol{f}}(\cdot)$ as training data is added when necessary. It implements a stochastic event-triggered adaptive control approach, which is highly data-efficient and reduces the number of costly measurements. In particular for mobile platforms this cost comes in terms of reduced battery life (processing data requires power for computations), shorter operation time (local data storage quickly fills up if operating near $1\,\mathrm{kHz}$ sampling rates) or increased communication efforts (in case data is not process or stored onboard).

The system (1) is inherently random due to the stochastic nature of the disturbance $\boldsymbol{f}(\cdot)$. Therefore, we combine feedback and feedforward control law

$$\boldsymbol{u} = \boldsymbol{K}\left(\boldsymbol{x} - \boldsymbol{x}^{\mathrm{des}}\right) + \boldsymbol{u}_{\mathrm{ff}}, \tag{8}$$

where $\boldsymbol{x}$, $\boldsymbol{x}^{\mathrm{des}}$ is the (desired) state of the quadcopter and $\boldsymbol{u}_{\mathrm{ff}}$ is a feedforward control term determined based on the known model $\boldsymbol{g}(\cdot, \cdot)$ (the gravitational force on the quadcopter) and the learned disturbance model $\hat{\boldsymbol{f}}(\cdot)$, more specifically the mean of the predicted disturbance $\mu(\cdot)$. The choice of the feedback gain $\boldsymbol{K}$, which compensates for imprecision in the model and the stochasticity of the disturbance, is a difficult trade-off because high control gains lead to high tracking performance, but also consume more energy and reinforce measurement noise, which can lead to instability. Therefore, it is generally advisable to let the feedforward term $\boldsymbol{u}_{\mathrm{ff}}$ take over most of the control effort and keep the feedback term small when the model is reliable. Therefore, we increase the gains only if the aleatoric uncertainty inferred by our model as $\sigma(\cdot)$ is high. Since the disturbance acts only in $z^q$ direction, only the $z$ component of the gain is affected i.e.,

$$k_z = \bar{k}_z\left(1 + \beta\sigma(\boldsymbol{x})\right), \tag{9}$$

where $\beta \in \mathbb{R}_{0,+}$ is the sensitivity and $\bar{k} \in \mathbb{R}_+$ defines the minimum control gain. This gain scheduling allows to robustify the closed-loop against process noise and can even guarantee stability Beckers et al. (2019), while at the same time we can keep the energy consumption low. While previous works by Fanger et al. (2016); Beckers et al. (2019) take a general uncertainty measure, we tune the feedback gains only based on the irreducible (aleatoric) uncertainty, while we combat the epistemic uncertainty with an increased data collection rate (7).

A summary of the control strategy is given in Algorithm 1 and a visualization of the neural network in Fig. 2. The outputs (red nodes) are iteratively trained with the indicated loss functions and respective data sets $\mathbb{D}$, $\mathbb{D}_{\mathrm{epi}}$ for two epochs with a a learning rate of 0.01 with each incoming data point. The current implementation pauses during the training of the disturbance model before the control loop continuous.

---

**Algorithm 1** Online learning controller

1: initialize disturbance model
2: **while** control task is not completed **do**
3:      get state $\boldsymbol{x}_i$
4:      evaluate disturbance model
5:      update control gains (9)
6:      **if** measurement is required (7) **then**
7:          measure disturbance $\boldsymbol{y}_i$
8:          update data set $\mathbb{D}_{\mathrm{tr}} \leftarrow \mathbb{D}_{\mathrm{tr}} \cup (\boldsymbol{x}_i, \boldsymbol{y}_i)$
9:          resample $\mathbb{D}_{\mathrm{epi}}$ (3)
10:         retrain disturbance model
11:      **end if**
12:      apply control $\boldsymbol{u}_k$ (8)
13:      i $\leftarrow$ i $+ 1$
14: **end while**

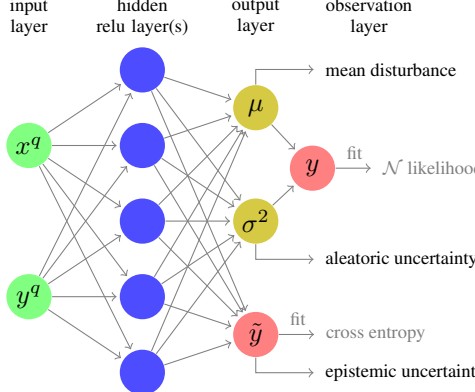

Figure 2: Schematic drawing of the NN to decompos the uncertainties for the quadcopter controller.

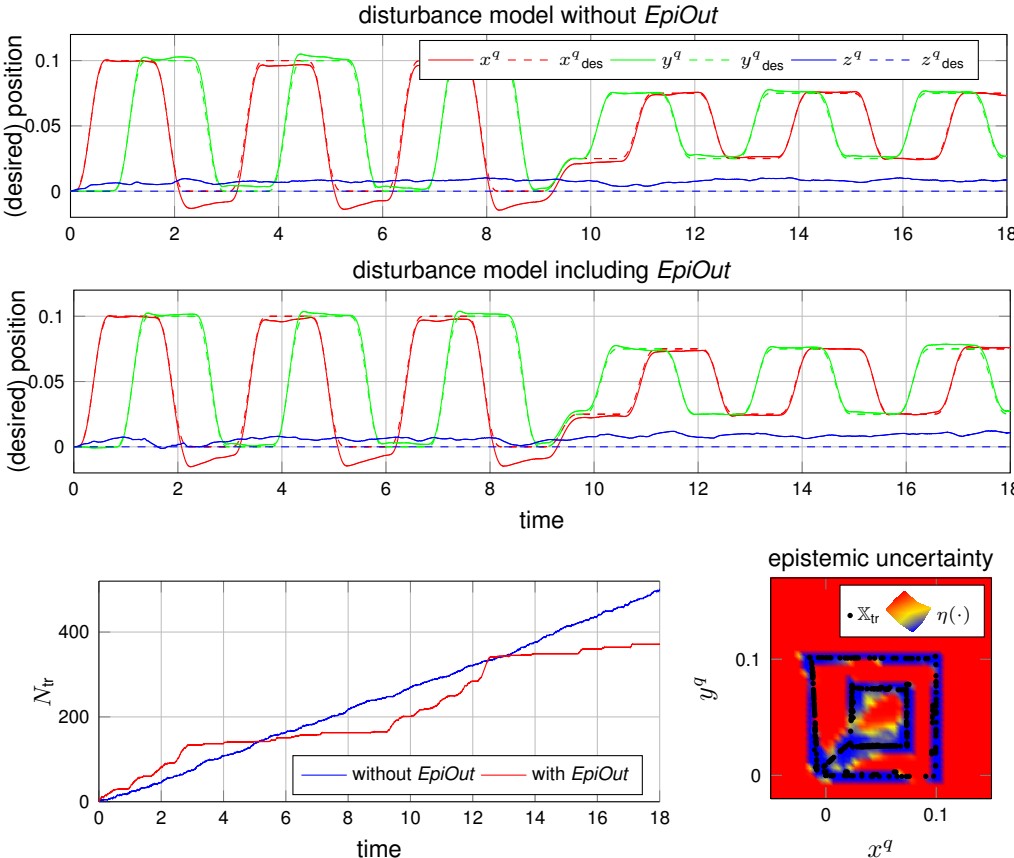

Figure 3: The *EpiOut* prediction of the disturbance model allows smart selection of the measurements taken. While a random selection constantly takes data points, out proposed approach takes more measurements when entering new areas ($0s$ to $3s$ and $9s$ to $12s$) and less when repeating the pattern (bottom). This results in overall improved tracking performance of a root mean squared error in z-direction of $0.00733$ (middle) vs $0.00773$ (top) while taking less data. The epistemic uncertainty estimate is low near training data and high in unobserved regions.

The tracking performance of the proposed quadcopter controller, the data collections rate and the epistemic uncertainty model is illustrated in Fig. 3. The implementation and further results are provided in the supplementary material.

## 5 CONCLUSION

This paper presents a novel deep learning structure for decomposing epistemic and aleatoric uncertainty and proposes an advanced control framework making distinct use of these uncertainty measures. As the prediction by the model are obtained sample-free, it allows for real-time critical online learning and outperforms existing methods on the proposed uncertainty-weighted precision measure. The proposed online learning control algorithm is inherently data-efficient by adding only required points to the data set.

For future work will consider alternative functions for sorting the epi points to encode prior knowledge, such as as periodicity (similar to a kernel of GP) and investigate the effect of a continuous valued $\tilde{y}$.

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
