# OpenReview forum: "Real-time Uncertainty Decomposition for Online Learning Control"
_ICLR.cc/2021/Conference — Reject_

### Official Review · AnonReviewer3 · 2020-10-26
**Unclear whether the method is principled. Update: review remains unchanged.**

**Rating:** 3
**Confidence:** 3

**Review:**

**Update**

My assessment remains unchanged. See the comments for details.
The proposed approach is circular---it tries to sample points at a "sufficient distance" from the data, then tries to learn this
"sufficient distance" by predicting the points. This circular dependency should be broken somehow. The current paper set this distance arbitrarily by sampling from a Gaussian with covariance $I$.

**Summary of work**

They propose an epistemic uncertainty estimation method (though it
is not really estimating the uncertainty, but just whether a datapoint
existed in the dataset or not). The proposed method samples new
points $x_{epi}$ in the vicinity of the training data points $x_{tr}$
from a Gaussian distribution (they set the covariance to $I$).
Then it creates a dataset where the target
is 1 for $x_{epi}$ and 0 for $x_{tr}$ (with some tweaks on removing N
datapoints from $x_{epi}$, which were the closest to training data points,
but the main principle is roughly the same). Then they train a network to
predict this target, and this estimate is called $\eta$.

They compare their new model against Gaussian processes, Bayesian neural
networks, Dropout uncertainty estimation on some datasets. To evaluate,
they propose a new weighted mean squared error metric, where the mean squared
error is weighted by $1-\eta$ and the mean is taken w.r.t. the sum of the
$1-\eta$ terms. For the other methods, $\eta$ is computed by dividing the
uncertainty for the prediction with the largest uncertainty in the data.
Their method outperformed the other methods using this metric, except for
GPs on some datasets, but they said that GPs do not scale, and are not
suitable for real-time applications.

They use these networks in a quadcopter simulation control task. Their
epistemic uncertainty estimation was used to decide whether to add a
data point to their data set (if the epistemic uncertainty is high) or
to discard it. The performance of their method including a disturbance
estimation model that uses their EpiOut approach improved the tracking
performance compared to not using any disturbance estimation model at all.

**Strengths and weaknesses**

Strengths:
- The exposition was good.

Weaknesses:
- It is not clear to me how to pick the covariance for the sampling distribution
of $x_{epi}$.
- It is not clear to me that there is a good justification for the method.
- It is not clear to me that using the largest uncertainty in the dataset
for computing the $\eta$ values for the other methods is appropriate. What if
the other method just predicts an infinite uncertainty somewhere? It seems
arbitrary to use the maximum, and it may be biasing the results to favor
the newly proposed approach. I have not seen this metric used anywhere else.
- The implementations use different frameworks, e.g. the GPs are based on
GPy using numpy (where there are other frameworks e.g. GPyTorch, Pyro,
or GPFlow that allow using GPUs), the BNN, dropout and EpiOut use tensorflow.
- The use in the quadcopter task was unclear to me. Why not just store all
of the data, or use each data point for a gradient update of the model, then
discard it?
- As far as I understood, in the experiment on the quadcopter the comparison
was between not using a disturbance model at all, and using a disturbance
model with EpiOut. It would be better to remove only the EpiOut component
to show that the uncertainty estimation was necessary, rather than just
any arbitrary disturbance estimation model.
- It seemed the number of data points gathered was around 150, which should
be easily handled with GPs, so the task does not seem to suggest that GPs
will not scale to it.
- The paper made up a new set of benchmarks for everything, and never tested
on an existing benchmark to prior works.

**Recommendation**

I recommend rejecting the paper with the main reason that I did not see
any principle behind the newly proposed method, and how the $x_{epi}$ dataset
could be constructed in a sensible way. Currently the covariance of the
Gaussians seemed to be set arbitrarily, but there should be a principled
way to set this covariance. Unless I have greatly misunderstood something,
I do not see how the paper could be modified for me to recommend it for
acceptance. I was also not convinced by the proposed quadcopter application for
the binary uncertainty estimation, and the other weaknesses listed above
also make me tend towards suggesting to reject.


**Questions**

Please clarify if there were misunderstandings.

**Additional feedback**

Below are some raw notes and additional thoughts from when I was reviewing
the paper.

"However, sampling the network during inference is a high computational
burden and is therefore not suitable in real-time critical control
tasks."
Really? With parallel computing it may not be that heavy. Do you have any
evidence for your claims? I see that in the appendix you have some
computational times, but it would have been good to point to this in the
main text.

"An extension to heteroscedastic GP regression is presented in
(Lazaro-Gredilla & Titsias, 2011), how-ever, it is a variational
approximation and further increases the computational complexity of
GPs,which is prohibitive when employing them for large data sets."
As far as I know, it is basically training two GPs, and the computational
complexity is the same, and differs only by a constant factor. Do you
have more evidence for your claim?

"preformed" -> performed

"considering a new data point takes $O(N_{tr}^3)$ computations"
No, I believe it takes $O(n^2)$ computation to add one new data point
using block matrix inversion.

Why was GPy used, and not some other framework with GPU support such as
GPyTorch, Pyro or GPFlow.

The Bayesian neural network was not clear, because it was a link to a
blogpost with many codes including numpy and tensorflow. (the code
cleared this up though)

"The measure in equation 6 relies on epistemic uncertainty prediction
in the interval[0, 1].  This is only ensured for the
proposed EpiOut approach and therefore the uncertainty measures provided
bythe GP, Dropout and BNN are scaled to the unit interval based on the
evaluation on all test points."
Is this a fair evaluation method? If one of the other methods gives an
infinite uncertainty on some point, this will negatively affect its entire
score, even though the uncertainty predictions for the other points may
be fine. The evaluation method does not appear sound to me, and no
references were provided for other works using such an evaluation method.

Being only confident close to the data trajectory may not be the best
approach. The model should be confident when the prediction is likely to
be accurate. For example if the model is smooth, then it should be
possible to interpolate reasonably accurately, and it should be OK
for the model to be confident as long as it has been able to learn
the structure sufficiently well.

---

> ### Author Response · Authors · 2020-11-20
> **Author response**
>
> We would like to thank the reviewer for the effort to evaluate and improve the submitted manuscript. Here are our responses to the raised concerns:
>
> **Responses to Weaknesses**
>
> *>> It is not clear to me how to pick the covariance..*
>
> We agree, that this is a crucial factor and not easy to choose. We want to make two important comments on this point:
> First, by scaling the input space to ensure that all training data points lie within $[0,1]^n$, the spread of the training data points Is standardized. This allowed us to use to same covariance on all evaluated data sets.
> Second, we must chose how far from data we want to trust our model (similar to GP SE lengthscales). If a Lipschitz constant is known, it can be used to tune the covariance of the Epi-samples: More spread out epi points lead to low epistemic uncertainty also outside of the training region, which holds for small Lipschitz constant, and vice versa.
> For the benchmarks, we did not want to use further knowledge and left the covariance at one.
>
>
> *>> It is not clear to me that  there is a good justification for the method. It is not clear to me that using the largest uncertainty...*
>
> We understand that this scaling requires special attention as concerns were also raised by another reviewer. In consequence introduceded a parameter $a \in [0,1]$ to scale $\eta$ optimally with respect to the performance measure. It shows no significant change on empirical evaluation.
>
>
> *>> The implementations use different frameworks...*
>
> That is correct, but we do not see a concern here, as the three approaches, which we actually compare with respect to computation time (*EpiOut*, *BNN*, *Dropout*) all operate on the same library.
>
>
> *>> The use in the quadcopter task was unclear to me...*
>
> First, thinking of the quadcopter as a time-continuous system, makes it difficult to decide what “all data” means as arbitrarily many data is available. Our event-triggered designs resolves this difficulty and is also popular in network control [1] and learning control [2].
> Second, data-efficiency becomes a necessity for most system with a sample rate in the kHz range, as storage on mobile systems must be considered to be limited.
> Third, we assumed that each measurement is associated with some cost, which is valid for the quadcopter: If the data is processed onboard (e.g. gradient updates at kHz range), battery life time is reduced. If the processing is not performed locally, the transmission puts high load on the communication channel.
> [1] Dimarogonas, Dimos, et al. "Distributed event-triggered control for multi-agent systems." IEEE TAC, 2011
> [2] Solowjow, Friedrich and Sebastian Trimpe. "Event-triggered learning." Automatica, 2020
>
>
> *>> As far as I understood, in the experiment on the quadcopter...*
>
> Thank you for this suggestion. We now compare it to the same model (without*EpiOut*), which gathers data uniformly. It shows a worse tracking performance while using more data points, because *EpiOut* triggers many measurements at the first round and benefits later from an improved model.
>
>
> *>> The paper made up a new set of benchmarks...*
>
> We assume this refers to eq. (6). We are not aware of a measure to evaluate the quality of the epistemic uncertainty and the regression. If any of the reviewers is familiar with such a measure, we would be happy to use it.
> We believe, that (6) is a quite natural choice under the assumption, that the epistemic uncertainty is bounded. One could argue, that it  grows infinitely as distance to the training set increases, but e.g. GPs with SE kernels also consider it to be bounded.
> Sarcos is – from our perspective - a well-established data set in the field of real-time regression for control, first published in [1] and utilized in [2,3,4] (among others).
> [1] Vijayakumar, Sethu, and Stefan Schaal. "Locally weighted projection regression: An o (n) algorithm for incremental real time learning in high dimensional space." ICML 2000.
> [2] Gijsberts, Arjan, and Giorgio Metta. "Real-time model learning using incremental sparse spectrum gaussian process regression." NN 2013
> [3] Meier, Franziska, et al. "Incremental local Gaussian regression." NeurIPS 2014
> [4] Andreas Doerr, et al. “Probabilistic Recurrent State-Space Models.“ ICML 2018
>
>
>
> **Responses to additional feedback**
>
> Of course parallelization can be of great help here. But for resource constraint (mobile) systems operating at khz this imposes a computational challenge.
> To avoid this misleading impression that heteroscedastic GPs scales worse, we have updated the cited statement in the paper.
> We now added to the paper that a block inversion can be achieved in quadratic time.
>
>
> *>> Being only confident close to the data trajectory may not be...*
>
> You are right, that additional knowledge should be used wherever possible and we are thinking of extensions, where the distance function $d$ (for sorting the epi points) is not simply an Euclidean distance, but- similar to a GP kernel – encodes prior knowledge.

---

> > ### Comment · AnonReviewer3 · 2020-11-21
> > **Still not convinced**
> >
> > Thank you for the response.
> >
> > I am glad that several of the experimental issues were improved e.g. adding an optimal weighting for the metric when evaluating the regression accuracy (although, I am a bit confused about these results: the EpiOut results seemed to change compared to the previous draft, and for BNN sometimes the weighted error increased when adding the optimal weighting) and performing the quadcopter task also with a disturbance estimation model while removing only the EpiOut component (In the new version the tracking performance was 0.00733 vs 0.00773, which is a much smaller increase in accuracy compared to before, and it was not clear to me that this is a practically relevant increase in accuracy).
> >
> > However, as I initially said it would be very difficult to convince me to recommend the paper.
> >
> > The main idea in the paper is to sample points away from the existing data points, then learn
> > a network to classify points that are far away from the existing data.
> > _This approach is circular: the first step of sampling the points at a sufficient
> > distance requires knowing what this sufficient distance is, yet the second
> > step learns a predictor for this sufficient distance from the sampled points._
> > The current paper just set this 'sufficient distance' arbitrarily to sample
> > from a Gaussian with covariance I. Yet, it is not explained why they do not
> > for example use 1e-10*I or 1e10*I instead. If they proposed a way to somehow
> > overcome this circular dependency, it may be an interesting paper to me, but
> > in the current state, I do not see how the work could be modified to be
> > acceptable to me. Finding a way to learn this sufficient distance is
> > the key part of estimating epistemic uncertainty, and it it is not
> > trivial. For example, Gaussian processes can maximize the
> > marginal likelihood to perform automatic relevance determination and
> > find the appropriate length scale parameters. Similarly, the current work
> > should provide a method to learn what is a suitable scale for the sampling
> > distribution, and what is a suitable metric d.
> >
> > In the rebuttal some additional comments were made regarding this distance, for example, a Lipschitz constant could be used, perhaps from some prior knowledge. Maybe there exist some sensible ways to find this constant based on heuristics. But I think the paper should discuss these points, and propose a method. Even if the data is rescaled to make the points lie between [0, 1], what matters is the distance across which the function that is being approximated would vary, and this is data dependent so can't be set arbitrarily. Moreover, if some methods are proposed to set this parameter based on prior knowledge, I believe it should be experimentally evaluated how sensitive the performance is to the setting of these length scales. Finally, it is not clear that a neural network would be necessary for evaluating the uncertainty: you could also try to just compute the distance to the nearest epi-point (this is done in the current paper using kd-trees anyhow), and use the distance to decide on whether the point should be added or not.
> >
> > Regarding metrics for estimating epistemic uncertainty, often one would just use the likelihood of the data; however, this is not applicable to your method, because you are not estimating epistemic uncertainty, but trying to give a binary estimate of whether the model is likely to be accurate or not.
> >
> > Finally, I think it would help if you can provide a specific real world example where your method would be useful, to make a stronger case that the method is relevant. The new arguments and references provided, e.g. battery time, communication costs, etc. are good arguments, but it is still difficult to say how relevant the method will be  without being explained a specific application. I understand that the authors are trying to create a method that is data-efficient and requires a low amount of computational resources, but it would help to know the specifications of a real world task (e.g. the amount of memory and computational resources on real currently used devices) to determine that the proposed method is indeed more suitable compared to alternatives. Moreover, it would be useful to explain in the paper that you are not considering the general epistemic uncertainty estimation task, but just predicting whether the uncertainty will be high or low, and explain why this would be useful (such explanation may be good in the introduction).

---

> > > ### Author Response · Authors · 2020-11-23
> > > **Thank you for the constructive feedback**
> > >
> > > Thank you for your constructive and detailed feedback. Here are a few thoughts based on your comments:
> > >
> > >
> > > *> Similarly, the current work should provide a method to learn what is a suitable scale for the sampling distribution...*
> > >
> > > You are right and we will work on an extension which picks up this idea: The norm of the gradient of the "regular" output of the NN (which is fitted to the training data with a mse or likelihood loss function) might be helpful in choosing the covariance.
> > >
> > >
> > > *> Finally, it is not clear that a neural network would be necessary for evaluating the uncertainty: you could also try to just compute the distance to the nearest epi-point*
> > >
> > > Similar to kernel density estimation methods, this would make the computational complexity not constant in the number of training points.
> > >
> > >
> > > *> It would help to know the specifications of a real world task*
> > > Thanks for pointing this out. We will do our best to set up an experiment, which actually includes these real-world constraints.

---

### Official Review · AnonReviewer2 · 2020-10-28
**Good paper, but questions are raised about the experiments.**

**Rating:** 7
**Confidence:** 4

**Review:**

Summary
--------
The authors consider the problem of efficient modeling of epistemic uncertainty, separated from aleatoric uncertainty,  for neural networks. They propose a novel methodology, involving automatically constructing a epistemic uncertainty support data set used to extend a given NN with an epistemic uncertainty output. The method is compared with previous, less efficient, approaches and is applied to the important problem of data-efficient online learning of a controller for real-time use with convincing results.

Strong points
-------------
1. The paper is well written, and the important problem considered is clearly presented and motivated.
2. The proposed approach with *epi points*, although seemingly simplistic at first glance due to its heavy dependence on a useful distance metric, is demonstrated to be quite effective for learning control in euclidean space.
3. The proposed method is demonstrated empirically in an online learning setting using real disturbance data, making the applicability convincing.


Weak points
-------------
1. While I recognizing that it is problematic to compare *bounded* (e.g probability value) and *unbounded* (e.g. regression target variance) epistemic uncertainty predictions, I wonder if isn't warranted to consider making the comparison fairer w.r.t. the baselines? Since the experimental maximum value for either baseline method depends on the choice of window size (i.e. $x\in [-4 , 4]$), a fairer comparison might be to find (optimize) a scaling parameter $\alpha \in [0,1]$ of the computed normalized epistemic uncertainty prediction, which minimizes (6). Basically a worst-case situation w.r.t. the proposed method, with the baselines performing the best they could have given any upper bound (larger than observed within the window) on their epistemic uncertainty prediction. I still expect that the proposed method will perform comparatively well based on for example the functional forms in Figure 1.
2. It is stated in the paper that "*... (Dropout and BNN) have a larger total discount on the training set than on the test set.*" (3.4, last bullet point). Can this be due to different maximum values between training and test when normalizing their epistemic uncertainty prediction?
3. I regard the inclusion of the Vanilla GP model baseline as important. A major concern, however, is about the treatment of this baseline in the experiment methodology:
   1. The epistemic uncertainty estimate for the GP is calculated based on the predictive variance. Is it not more appropriate to use the predictive standard deviation instead, such that the uncertainty has the same unit as the target? (The same applies to Dropout and BNN too?)
   2. The data in *1D center* and *1D split* is very well suited for GP regression with an SE ARD kernel and should report a well calibrated epistemic regression uncertainty in input ranges where training data is absent. I find the miss-match between the left figure (showing the the GP mean prediction deviation) and the right figure (showing the epistemic uncertainty estimate) in Figure 1 odd. Implementing 3.1. might alleviate this issue?
   3. I find it odd that EpiOut outperform the GP model on *1D center* and *1D split*, taking into consideration the performance metric (6). Maybe it is a consequence of issue 3.1.?
   4. Table 4 and Table 5 in the appendix: It is odd that the GP test discount is smaller than the GP training discount on 2D Gaussian. It would indicate, as mentioned in the paper, that the epistemic uncertainty estimate is off for the GP model. Figure 3 (appendix) does indeed seem odd (or unclear) for the GP model, as does Figure 2 (appendix). Maybe it is a consequence of issue 3.1.?
4. I am missing details on the online learning for the experiment, e.g: How long is each NN training and how is it related to the real-time aspects? Will the quadcopter fly far into state for which the current NN designates as high epistemic uncertainty before the NN is updated with new data points and  retrained such that the epistemic uncertainty is reduced? Or is the simulation paused and waiting for the NN training to complete on every new data point? If so this should be made clear in the paper.

Reason for score
----------------
It is a good paper on an important topic, with supportive empirical evaluation of the applicability of the proposed novel approach. My major concern is with certain aspects of the comparative evaluation and about the clarity of the online learning experiment.

Minor comments
--------------
1. I suggest that references to material in the appendix are made in the paper. Technicalities such as the online learning algorithm and relevant results would be made clearer.
2. Figure 4: Left and middle figures are a bit hard to interpret. It seems like the figures show multiple runs, which makes them even more difficult. They might benefit from a thinner marker/stroke, transparency and/or a 3D effect such as a color gradient for view depth etc. They could also be changed to plotted against time instead of against space.
3. GP regression with a squared exponential (SE) automatic relevance determination (ARD) kernel has, for a given training data (X,Y), a maximum epistemic regression uncertainty, given by the signal variance parameter $\sigma_f$ of the kernel: As $d(x^*, x)$ grows the exponential in the kernel decrease towards 1 rapidly, where $x$ is a training data point and $x^*$ is an arbitrary data point. The epistemic uncertainty is consequently bounded for the selected choice of kernel family.

3.5, bullet list, first bullet:
  "\textit{EpiOut}model" -> "\textit{EpiOut} model"
  "\textit{EpiOut}is" -> "\textit{EpiOut} is"

4.1, paragraph 1:
  "\textit{EpiOut}approach" -> "\textit{EpiOut} approach"
  "\textit{EpiOut}\eta(\cdot)" -> "\textit{EpiOut} \eta(\cdot)"

Edit: Markdown problem with nested lists

Edit: Upgraded rating due paper improvements

------------
Thank you for addressing my concerns.

Figure 1 is not updated (it looks the same w.r.t. the GP, but the change from variance to std should be noticeable?).

---

> ### Author Response · Authors · 2020-11-20
> **Author response**
>
> We would like to thank the reviewer for the effort to evaluate and improve the submitted manuscript. Here are our responses to the raised concerns:
>
> Responses to Weak points
> --------------------------------------
> **Answer to (1)**
> Thanks for providing us with this improved idea for the evaluation. We have implemented it and now report the updated results, which have - as you already guessed – not changed significantly.
>
> **Answer to (2)**
> Yes, you are right. On the initially presented data, we considered training and test data separately from each other and therefore the normalization was not harmonized. Rerunning the evaluation with the same normalization shows that now also BNN and Dropout show consistent results, i.e. lower average epistemic uncertainty on the training than on the test set. Thank you for pointing this out. We have updated the numbers accordingly and removed the cited statement.
> Nevertheless, we would like to  emphasize, that the key performance measure, the weighted mean square error on the test set, is not affected by this change, as it concerns mainly the predicted epistemic uncertainty on the training data set. Furthermore, this different method of evaluation further underlines the difficulty to rank a epistemic uncertainty prediction as “high” or “low” if it is not normalized.
>
> **Answer to (3)**
> For *Dropout* and *BNN* we are considering the predictive standard deviation and now also consider it for the *GP model*. We agree that standard deviation is a better measure due to the arguments you have brought forward and will state this more clearly in a future revision of the paper. Nevertheless, we have not seen a significant difference in the empiric evaluation.
>
> **Answer to (4)**
> We are afraid, we do not understand which miss-match you are pointing out here. From our perspective, the *GP model* fits the true function perfectly for $|x|<3$ where it also predicts a low epistemic uncertainty, vice versa for $|x|>3$. For $|x|<-1$ one might expect a larger epistemic uncertainty estimate due to missing data but does not become a problem as long as the mean prediction is also accurate in this area.
>
> **Answer to (5)**
> We have reconfirmed the results with the posterior standard deviation of the GP, so it is not a consequence of the provided issue in 3.1. While the GP shows – from our perspective - the desired behavior in terms of the epistemic uncertainty estimate, it is not guaranteed to be optimal with respect to the performance metric in (6). Therefore, we do not see a problem in outperforming the GP as it is not a baseline in a sense that it achieves the best possible result here.
>
> **Answer to (6)**
> You are right here, that this inconsistency is resolved when scaling the epistemic uncertainty for the training and the test set in the same way. Thus, it is not a consequence of 3.1 but of 2.
>
> **Answer to (7)**
> Around the training of the NN there are a lot of design choice being made. Currently, the NN is retrained with every incoming data point for (2 epochs, learning rate 0.01). But we tested it also if it is retrained every n-th data point with more epochs per training. While it leads to less frequent training, it accumulates more training data points, because high epistemic uncertainty is predicted by the outdated model in the proximity of measured (but not trained) data points.
> At the current state, the simulation is paused until the training is complete, but our future work aims towards an asynchronous implementation, where the training is performed in background and an outdated model is used until training completes. Such a design has already been used in [1].
> To make this clear to the reader, we have now provided a pseudo code in the main paper.
> More important with respect to the real-time capability was to make the epistemic uncertainty prediction fast because this is performed at a much higher rate than the training.
>
> [1] Lutter, Michael, Christian Ritter, and Jan Peters. "Deep Lagrangian networks: Using physics as model prior for deep learning." International Conference on Learning Representations (ICLR)
>
> Responses to Minor comments
> -------------------------------------------
> **Answer to (1)**
> Thanks for the advice. We have moved the algorithm to the main paper and point to the supplementary material where necessary.
>
> **Answer to (2)**
> The figure shows only a single run, but each run consists of 3 labs/rounds. We have now stated the desired trajectory explicitly. Furthermore, based on your other comments, we decided to move this particular plots to the supplementary material to make space for more insightful graphics.
>
> **Answer to (3)**
> Yes, we agree. If we have stated something conflicting in the paper, please let us know and we will correct it.
>
> Thanks for pointing out the typos. We have updated the manuscript accordingly.

---

### Official Review · AnonReviewer4 · 2020-11-05
**Epistemic uncertainty is output from NN using data sampled outside training data**

**Rating:** 6
**Confidence:** 3

**Review:**

Epistemic uncertainty is a useful measure to have from a learned model. This paper proposes to output epistemic uncertainty from NN by training with samples outside the training set. My concerns are listed below:

1. The paper samples epi points around the training set. How does this perform when training set is very sparse or very spread out.
2. Changing the yellow line to something darker and also making the thick black line to be a lighter color would improve clarity of the results.

---

> ### Author Response · Authors · 2020-11-20
> **Author response**
>
> We would like to thank the reviewer for the effort to evaluate and improve the submitted manuscript. Here are our responses to the raised concerns:
>
> **Answer to (1)** As we sample around each training point individually, we specifically account for spread out data. To evaluate this empirically, we created a new dataset with only 2 training data points (labelled as 1D sparse) and *EpiOut* still shows meaningful epistemic uncertainty estimates (as opposed to *BNN* or *Dropout*) as shown in the updated supplementary material.
>
> **Answer to (2)** Thanks for the advice. We have updated the illustrations accordingly.

---

> > ### Comment · AnonReviewer4 · 2020-11-20
> > **The sparse experiment is helpful**
> >
> > Thank you for running the additional experiment on sparse data and for your clarifications.

---

### Official Review · AnonReviewer5 · 2020-11-05
**An interesting paper about uncertainty quantification in dynamics learning for control. The proposed method EpiOut is sample-free at the inference time and it outperms some baselines. But there are no theoretical justification or insights, and the binary epistemic output is too simplified.**

**Rating:** 5
**Confidence:** 3

**Review:**

**Pros and the Key Idea**
This paper studies uncertainty quantification (UQ) in model-based learning for control, which is a timely and important research direction. The proposed method (EpiOpt) trains a neural network to predict the epistemic uncertainty directly. The training data for epistemic uncertainty prediction is artificially generated based on a simple nearest neighbor principle. The key ideas are: given the labeled training dataset $X_{tr},Y_{tr}$ (the data size $|X_{tr}|=N_{tr}$), this paper first randomly samples $X_{epi}$ around $X_{tr}$, where $|X_{epi}|=N_{epi}=k\times N_{tr}$. Then this paper labels $x\in X_{epi}$ by $1$ if the minimum distance from $x$ to $X_{tr}$ is far, and by $0$ if the distance is short. Finally, a neural network is trained for this binary classification task.

Finally, this paper uses this idea in online control: the learned epistemic uncertainty is for adaptive data collection, and the aleatoric uncertainty is for control gain tunning. The advantage of this framework is that it is very simple, and doesn't need sampling or test-time dropout at the inference time.

**Cons and Suggestions**
(1) Many related work is missing especially for domain shift and adaptive control. As mentioned in this paper, the epistemic uncertainty is mainly from the data distribution shift, but there is no discussion about domain shift in this paper. The main idea of domain shift in ML is to *quantify the "distance'' between the source and target domains*, which is similar to the epistemic uncertainty prediction $\eta(\cdot)$ in this paper. People also considers domain shift in control and learning (http://proceedings.mlr.press/v120/liu20a.html, https://arxiv.org/abs/2006.13916 and many others).
Also, adaptive control can handles epistemic uncertainty in an online manner as well. It would be great to discuss the difference between adaptive control.
The ensemble method (e.g., https://proceedings.neurips.cc/paper/2017/file/9ef2ed4b7fd2c810847ffa5fa85bce38-Paper.pdf) is also an active method for UQ.

(2) The key method, *EpiOpt*, is a bit too simple and there are no theoretical justifications or insights, such that it is hard to convince me about the generalirity and robustness of this method. As I mentioned above, the key idea of *EpiOpt* is to train a "distance function" $\eta(\cdot)$ to quantify the distance between the source and target data. This idea has been both theoretically and empiracally studied in domain shift and transfer learning. A few questions pop up from this paper:
* Why do you need to sample $X_{epi}$ around $X_{tr}$? Since the goal is to get $\eta(\cdot)$, why don't you consider some analytical solutions such as KDE (kernel density estimation) to derive $\eta(\cdot)$? In other words, you could just use $X_{tr}$ to estimate a density function for the source data, and then evaluating this density function to get $\eta(\cdot)$. I didn't see a clear reason to *train* a neural network to estimation this distance.
* In equation (5), this paper labels $X_{epi}$ either $1$ or $0$. Why isn't it a continuous value from $0$ to $1$? For example, you can rank $d_j$ to get this continuous value.

(3) The title and abstract of this paper emphasize a lot on *epistemic and aleatoric uncertainty decomposition*. However, the key method *EpiOpt* is only about the epistemic uncertainty, and how to deal with the aleatoric uncertainty only appears in the experimental section in equation (9). I highly recommend the authors discuss about these two types in the method section (Section 3) and give a general framework. The current aleatoric uncertainty is more like a gain tunning method, which is not related to *uncertainty decomposition* .

(4) The experimental section is a bit vague. I highly recoomend the authors present the concrete learning and control problem first, e.g., Which part in the dynamics is learned? How to collect data? How to decompose the epistemic and aleatoric parts? A good example is https://ieeexplore.ieee.org/abstract/document/8794351, where the task is very similar.

**Code-of-Ethics**
I see no ethic issues in this paper.

---

> ### Author Response · Authors · 2020-11-20
> **Author response**
>
> We would like to thank the reviewer for the effort to evaluate and improve the submitted manuscript. Here are our responses to the raised concerns:
>
> **Answer to (1)**
> Thank you for pointing us to further relevant literature. We agree that our epistemic uncertainty estimate corresponds to the domain shift as a quantification between the source and target domain. We therefore added the suggested work [1] on domain shift in control to our literature review. It has a slightly different focus as the desired trajectory can be actively chosen for exploration and guarantees safety, while our work passively follows a given trajectory.
> We rather see our work in line with adaptive control or also dual control, which considers the task of simultaneously controlling and identifying an unknown system [2] and we have added further references to our literature review. The key difference we see is that adaptive control typically works on a time-triggered base while our adaptation is event-triggered, which is only used in the minority of previous work and shows promising results, e.g. in [3] or [4].
> The mentioned work on ensemble methods [5] uses a similar technique as MC-dropouts as it also preforms predictions at the same inputs with all members of the ensemble to obtain a distribution over the output. Due to the low number of nets in the ensemble, this might be possible in a real-time constrained situation but the approach in does not differentiate between aleatoric and epistemic uncertainty. We will include it in the related work of the paper.
> [1] Liu, Anqi, et al. "Robust regression for safe exploration in control." Learning for Dynamics and Control. 2020.
> [2] Wittenmark, Björn. "Adaptive dual control methods: An overview." Adaptive Systems in Control and Signal Processing. 1995
> [3] Solowjow, Friedrich, and Sebastian Trimpe. "Event-triggered learning." Automatica. 2020.
> [4] Umlauft, Jonas, and Sandra Hirche. "Feedback linearization based on Gaussian processes with event-triggered online learning." IEEE Transactions on Automatic Control (2019).
> [5] Lakshminarayanan, Balaji,  et al. "Simple and scalable predictive uncertainty estimation using deep ensembles." Advances in neural information processing systems. 2017.
>
> **Answer to (2)**
> Thank you for drawing our attention to KDE as alternative approach. Like GPs, kernel density estimation is a non-parametric approximator, i.e. the computational effort of the prediction grows with the number of data points. Since the epistemic uncertainty prediction in our control application is used to determine whether a new data point is accepted or not, it is evaluated quite often (compared to a training of the model). The computational effort for prediction with *EpiOut* is constant in the number of training data points. We understand that this requires further explanation and will discuss KDE in the related work.
> The idea to employ continuous values from 0 to 1 for $\tilde{y}$ sounds interesting and we will investigate its effect in the future. We have pointed out this idea in “future work”.
>
> **Answer to (3)**
> Thank you for pointing this out. As we see one main contribution in the *EpiOut* algorithm and the other in the control design, we decided that the aleatoric uncertainty estimation should not have such a prominent position (as it also has limited novelty). However, we now illustrate the NN structure in the experimental section and hope that this puts more focus on the uncertainty decomposition. We have also rewritten parts of the introduction and the problem formulation to adjust the structure based on your comment.
> The uncertainty decomposition is a prerequisite for the gain tuning method as it is based on the aleatoric uncertainty only.
>
> **Answer to (4)**
> Thank you for the great advice. To visualize the uncertainty decomposition, we have added a schematic drawing of the employed neural network and also present the pseudo code showing the data collection strategy.
> Regarding the specific dynamics, we have added a citation to the given resource [1] since we consider the ‘’classical’’ quadcopter dynamics. We made it now more clearly, that these dynamics are known to the controller denoted as $g(\cdot,\cdot)$ in our problem formulation. The unknown part is the disturbance given by the thermals, which our model estimates. To clarify these, we have completely restructured Section 4.
> [1] Shi, Guanya, et al. "Neural lander: Stable drone landing control using learned dynamics." International Conference on Robotics and Automation (ICRA). 2019.

---

### Decision · Program_Chairs · 2021-01-07
**Final Decision**

**Decision:**

Reject

**Comment:**

This paper aims to do efficient epistemic uncertainty quantification for model-based learning for control. It does so by augmenting the dataset with synthetic data around the true data points, and trying to classify whether a point is close to the training set or not. I agree with many of the criticisms that R3 and R5 brought fourth. Namely, it's not clear why a kernel density estimate couldn't be used instead (runtime complexity is cited as the reason, but could be addressed through approximations, inducing points etc). It is not clear how to set the sampling distribution for X_epi. Also, since efficiency is a motivation for the work, I suggest that the authors look at and cite:

https://arxiv.org/abs/2002.06715

I think at the moment the paper is not ready for publication, but the idea is interesting. Aside from comparing with the work above, what would improve this paper is an automatic way to select the distribution, or at least the covariance, of X_epi.